# Mechanistic Investigation of Typical Loess Landslide Disasters in Ili Basin, Xinjiang, China

**Maoguo Zhuang** [1,2,3,4], **Wenwei Gao** [5,*] , **Tianjie Zhao** [6], **Ruilin Hu** [1,2,3], **Yunjie Wei** [4], **Hai Shao** [4] and **Sainan Zhu** [4]

[1]    Key Laboratory of Shale Gas and Geoengineering, Institute of Geology and Geophysics,
       Chinese Academy of Sciences, Beijing 100029, China; jcyzhuangmaoguo@mail.cgs.gov.cn (M.Z.);
       hurl@mail.iggcas.ac.cn (R.H.)
[2]    Institutions of Earth Science, Chinese Academy of Sciences, Beijing 100029, China
[3]    College of Earth and Planetary Sciences, University of Chinese Academy of Sciences, Beijing 100049, China
[4]    China Geological Environment Monitoring Institute, Beijing 100081, China;
       jcyweiyunjie@mail.cgs.gov.cn (Y.W.); jcyshaohai@mail.cgs.gov.cn (H.S.); jcyzhusainan@mail.cgs.gov.cn (S.Z.)
[5]    Institute of Architectural Engineering, Yan'an University, Yan'an 716000, China
[6]    Aerospace Information Research Institute, Chinese Academy of Sciences, Beijing 100101, China;
       zhaotj@aircas.ac.cn
*    Correspondence: gaowenwei@yau.edu.cn

**Abstract:** In the period from 2010 to 2018, a total of 302 geological disasters occurred in the Xinjiang Autonomous Region, China, of which 136 occurred in the Ili Basin. Compared with those in other regions, the loess landslides in the Ili Basin are strongly influenced by the seasonal freeze–thaw effect. Taking the No. 2 Piliqinghe landslide as an example and based on the field geological investigation, it was found in the present study that the main triggering factors of this landslide were the snowmelt erosion of the slope toe and meltwater infiltration into the trailing edge of the slope. The mechanism of loess landslide instability was studied using numerical simulation. The results showed that (1) the Piliqinghe landslide disaster was formed through a process composed of the local sliding of the leading edge → the creep sliding and tension cracking of the slope surface → the overall sliding stage; (2) the infiltration of snowmelt was the direct cause of the landslide formation; (3) the fluvial erosion and softening caused the soil of the slope toe to slide. The results can be used as a reference for the analysis of the disaster mechanism and movement characteristics of similar loess landslides.

**Keywords:** loess landslide; freeze–thaw effect; disaster mechanism; numerical simulation

## 1. Introduction

Loess is a special Quaternary loose sediment deposited and consolidated in a special natural environment [1]. Due to the special structural and physico-mechanical properties of loess, loess landslides have occurred frequently in the world, such as the loess landslides in the Kyrgyz and TajikTien Shan (e.g., the Kainama earthflow, the Shrara flowslide, the Firma flowslide, the 1 May flowslide, etc.) [2,3]. In China, loess is distributed over an area of $64 \times 104$ km$^2$, accounting for 6.3% of the total area of the country [4]. Statistically, excluding earthquake-induced landslides, approximately one-third of all landslides in China occurred in loess areas [5,6]. In recent years, the intensification of human economic and engineering activities have been accompanied by the increasing occurrence of loess landslide disasters, posing a serious threat to people's lives and property and even causing mass injuries and casualties. Research on the triggering factors and formation mechanisms of loess landslides is the basis for effective mitigation of the risk of these landslides [7–9]. Since the 1980s, Chinese researchers have been investigating loess landslides and proposed a number of mechanisms, including those for the movement and liquefaction of saturated sand–gravel layers of the sliding bed in high-speed long-runout landslides, earthquake-induced liquefaction and pulverization effects, and static liquefaction and flow liquefaction [10–12].

Rainfall, irrigation, earthquakes, and human activities are considered the main factors triggering loess landslides [13–18]. For example, in August 1955, a loess landslide with a volume of $3 \times 10^7$ m$^3$ occurred in Wolongsi of Baoji, Shaanxi Province, destroying a railroad, and a landslide occurred again at this location in 1971, killing 28 people [19]. On 7 March 1983, a landslide with a volume of approximately $4.1 \times 10^7$ m$^3$ occurred in Dongxiang, Gansu Province, causing the collapse of more than 500 houses and the death of 237 people [20]. In 1991, a massive landslide occurred in Tianshui, Gansu Province, killing 200 people [21]. On 15 March 2019, a loess landslide occurred in Zaoling, Shanxi Province, causing the collapse of multiple houses and killing 20 people [4]. These catastrophic events have greatly raised the public awareness of loess geological disasters and promoted special research on loess landslide-triggering factors.

The role of seasonal freeze–thaw cycles is often overlooked among the many landslide-triggering factors such as rainfall, climate, earthquakes and human activities [22]. However, in the arid and semiarid regions of Northwest China, freeze–thaw-induced landslides occur frequently, causing significant human injuries and property damage [23]. For example, on 10 March 2010, the Zizhou landslide in Yulin, Shaanxi Province, killed 27 people; and on 7 February 2012, the Jiaojia landslide in Yongjing, Gansu Province, pushed two vehicles into the Yellow River, killing one and leaving three people missing [24]. In recent years, the continuous occurrence of geological disasters during the freeze–thaw period has attracted increasing attention from the public and researchers. Studies on the mechanisms of freeze–thaw-induced landslides can be divided into two categories: one focuses on the destruction of the soil structure caused by freeze–thaw cycles, leading to a decrease in the physico-mechanical parameters of the soil and thereby triggering landslides, and the other pays attention to the water table rise due to freezing, which results in an increase in the pore water pressure and thus triggers landslides [25,26].

Although the Xinjiang region of China is not a high-risk area for geological disasters, a relatively large number of loess landslides occur in the Ili region of Xinjiang. A study of the vertical zone spectrum of natural disasters in the Tianshan Mountains region revealed that snowmelt and rainfall in spring are usually the main factors triggering landslides [27]. The frequent occurrence of landslides in the Ili region during the winter–spring transition period makes the investigation of the seasonal freeze–thaw effect the first choice. To further improve the accuracy of disaster assessment for this region, researchers investigated the formation mechanism and dynamics of the 31 July 2012, catastrophic landslide in Ili and found that the loess landslides in this region are characterized by structural complexity and instability due to unsaturated rainfall seepage. In particular, the freeze–thaw cycle leads to the enrichment and dissipation of groundwater in the slope. Enrichment results in the softening of the soil in the loess slope, while dissipation causes abrupt changes in the hydrostatic and hydrodynamic pressures, which simultaneously changes the pore characteristics and permeability, hence affecting the stability of the loess slope [28,29]. Sudden heavy rainfall is an important factor triggering loess landslides in the Ili region [30,31]; therefore, the study of loess landslide in the Ili region for the seasonal freeze–thaw effect loess landslide is typical and representative.

In this paper, taking the No. 2 Piliqinghe landslide as an example and based on a field geological survey, the characteristics and disaster mechanism of this landslide are analyzed in depth, and the disaster mode of this type of landslide is summarized to provide scientific experience and a theoretical basis for the rational disaster prevention and mitigation of and emergency responses to landslides in the loess region of Xinjiang, China.

## 2. Geological Conditions of the Study Area and Research Methods

### 2.1. Geological Conditions

The Ili Basin is located in the western part of the Tianshan Mountains in Xinjiang, with longitude in the range of $80°10'$ to $84°58'$ E and latitude in the range of $42°20'$ to $44°49'$ N. Geological disasters occur frequently in this region due to natural factors such as a complex geological structure, rock fragmentation, and unique hydrometeorology as

well as intense human activities. According to the survey data of the China Geological Environment Monitoring Institute, a total of 302 geological disasters occurred in Xinjiang from 2010 to 2018, of which 136 occurred in the Ili Basin (Figure 1), accounting for 45.03%, with landslides as the main type of geological disasters but fewer debris flows and collapse.

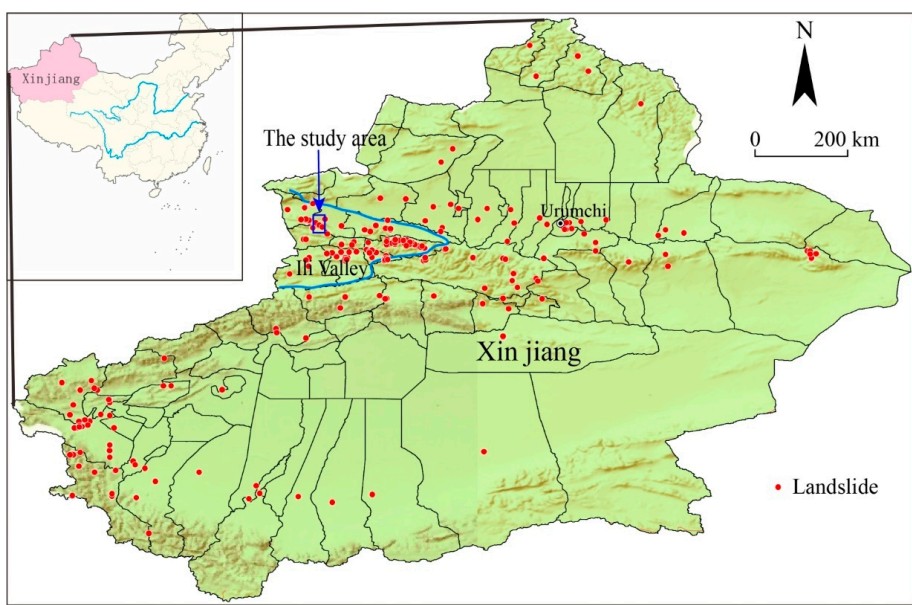

**Figure 1.** Distribution of landslide disasters in the Ili River Valley region of Xinjiang.

The Ili Basin has a complex topography and geomorphology, presenting an overall pattern of "three mountains sandwiching two valleys and one basin". The geomorphological types are divided into three primary geomorphological units, namely, fault-block mountains with eroded folds, block uplift mountains by denudation and accumulation, and accumulation plains. The geological disasters in the region mostly develop in piedmont loess hills, which are distributed in the piedmont area and belong to block uplift mountains by erosion accumulation, with elevations of 900–1600 m and relative heights of 50–100 m. The hills are banded and were formed by uplift due to neotectonic movement. The Meso-Cenozoic strata are often covered with gravel and loess as well as lush vegetation, and the local lowlands are prone to landslides due to spring outcrops. The loess strata are mostly distributed in the foothills on two sides of the valleys or cover the slopes of the bedrock in the mountainous area, with the upper boundary reaching the forest belt and the lower boundary meeting the river valley plains. Their thickness is generally approximately 5–30 m and can reach a maximum of 80 m at some individual locations.

The Ili Basin is located in a high seismicity zone with intense neotectonic movements that mainly manifest as the inherited elevation and subsidence and the accompanying faults and folds. This basin is located in the North Tianshan seismic zone and immediately adjacent to the South Tianshan seismic zone in the south. Both seismic zones compose the main high seismicity regions in Xinjiang. Since 1812, there have been 14 strong earthquakes with magnitudes greater than six in the Ili Basin and adjacent regions.

The Ili Basin is situated in the center of the Eurasian continent and in western Xinjiang, which is far from the ocean, thereby forming a typical continental climate with an average annual temperature of 7.8 °C and an average annual precipitation of 349.3 mm. In a year, the temperature rises rapidly in spring, the summer is mild and rainy, the temperature drops quickly in the autumn, and the winter is long and warm (Figure 2). The supply of surface water is clearly controlled by seasonal precipitation and influenced by both alpine snowmelt and precipitation recharge (Figure 3). Geological disasters in the region are often distributed along both sides of the valleys, and perennial flowing water has a direct impact on geological disasters.

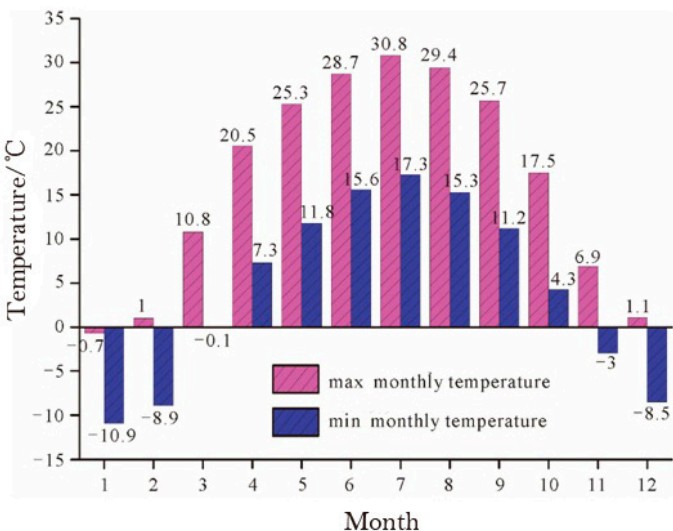

**Figure 2.** Monthly temperature in Ili between 2010 and 2018.

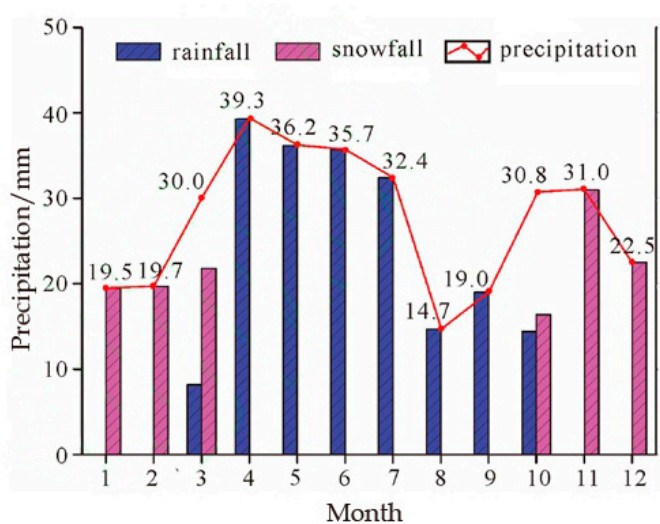

**Figure 3.** Monthly precipitation in Ili between 2010 and 2018.

The mountainous area in the Ili Basin accounts for more than 70% of the total area of the region, with a mild climate and abundant precipitation. Human engineering and economic activities are mainly concentrated in the valley plain area and in the fields of, for example, agricultural and hydraulic engineering, animal husbandry, and the tourism industry.

### 2.2. Methods

Mechanistic analysis of landslide instability is the core issue in landslide disaster research. Due to the complexity of geological conditions and the limited understanding of special landslides, the field investigation and analysis of engineering geological conditions are of great importance in landslide research. The qualitative analysis of engineering geology is especially valuable for research on landslide disasters with complex geological conditions and diverse triggering factors.

Stability under special conditions is another core issue in landslide disaster research. At present, the numerical simulation method is commonly employed. When heterogeneous and nonlinear slopes with complex boundaries are studied with this method, computer-based numerical software is used to obtain the stress–strain relations of rock and soil and to quantitatively analyze the slope deformation/damage and groundwater seepage to investigate the interaction between the rock and soil. The complex stratigraphic conditions

of the slope can be simulated in a cost-effective and efficient manner to obtain the stability state of a landslide under different working conditions.

The integration of field investigation, qualitative engineering geological analysis, and quantitative numerical simulation is the most common and most comprehensive approach in the study of landslide mechanisms. The integrated approach makes it possible to reveal the fundamental causes and formation modes of landslides and provide theoretical support for disaster prevention and mitigation.

A schematic diagram of the landslides research methods of this study is detailed in Figure 4. First, we research the landslides in the slope region of the Ili Basin in Xinjiang using remote sensing image analysis and geological investigation. Then, we build a geological structural model of the landslide and simulate its deformation and failure process using the numerical software FLAC [3D] 5.00. Finally, we discuss the loess landslide's stability, deformation and failure mechanism.

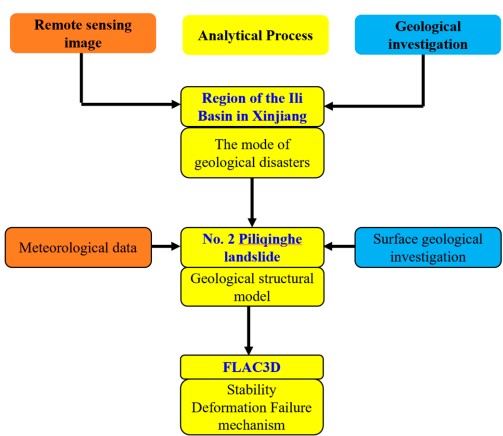

**Figure 4.** Schematic diagram of the landslides research methods.

## 3. Case Study of Typical Geological Disasters in the Study Area and their Disaster Modes

Geological disasters in Ili between 2010 and 2018 are shown in Table 1. Geological disasters occurred mainly in spring and summer in the Ili Basin, with 92 (67.65%) in spring, 40 (29.41%) in summer, 4 (2.94%) in autumn, and none in winter. The Ili Valley is a region with extensive development of freeze–thaw landslides. Past records show that 152 landslides occurred during the freeze–thaw periods, accounting for 40% of the 380 landslides recorded in the Ili Valley region [31,32]. According to the spatiotemporal distribution characteristics of loess landslides in the Ili Basin, the main triggering factors for loess landslides in this region are precipitation, fluvial erosion, and freeze–thaw cycles [33,34].

**Table 1.** Geological disasters in Ili between 2010 and 2018.

| Year | Total Number in Xinjiang | Ili | | | |
| | | Total Number | Spring (March to May) | Summer (June to August) | Autumn (September to November) | Winter (December to next February) |
|---|---|---|---|---|---|---|
| 2010 | 68 | 31 | 31 | | | |
| 2011 | 11 | 6 | 3 | 1 | 2 | |
| 2012 | 40 | 6 | 4 | 1 | 1 | |
| 2013 | 29 | 6 | 1 | 5 | | |
| 2014 | 15 | 2 | 2 | | | |
| 2015 | 13 | 4 | 2 | 2 | | |
| 2016 | 65 | 35 | 17 | 18 | | |
| 2017 | 61 | 40 | 29 | 11 | | |
| 2018 | 28 | 6 | 3 | 2 | 1 | |
| Sum | 330 | 136 | 92 | 40 | 4 | |

### 3.1. Piliqinghe Landslide

Located in the west of the Ili Valley, the Piliqinghe River is a typical area in the Ili region that experiences loess landslide disasters. As shown in the remote sensing image of the Kezilesai Gully (Figure 5), the survey found that there were 14 loess landslides in the Piliqinghe River Basin, all of which developed in the three first-level tributaries in the upper reaches of the Piliqinghe River. Among them, the third branch gully (the Kezilesai Gully in Figure 6, the remote sensing image was taken by UAV(Unmanned Aerial Vehicle) with a resolution of 50 cm), which is relatively small, developed nine landslides (No. 1 to 9), with the leading edges of two landslides located on floodplains; the branch gully on the right side of the two main branches is the Piliqinghe River, which developed five landslides (No. 10 to 14), with the leading edges of four landslides located on first-level terraces or floodplains; and the branch gully on the left side is the Axi Gully, which developed four landslides (nos. 15 to 18), with the leading edges of two landslides located on first-order terraces or river floodplains.

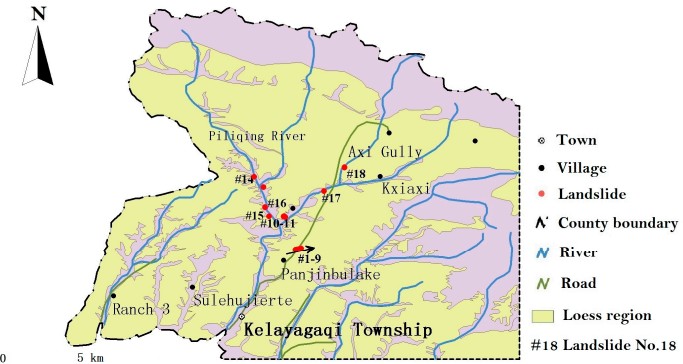

**Figure 5.** Distribution of the Piliqinghe landslides.

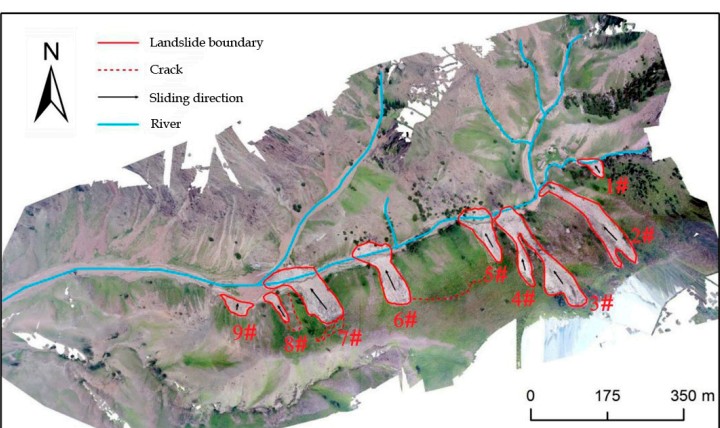

**Figure 6.** Remote sensing image of the Kezilesai Gully landslides in the Piliqinghe basin.

The Kezilesai Gully landslides in the Piliqinghe Basin are shown in Figure 7; the Pili qinghe landslide is a homogeneous loess landslide of a single lithology. The Ili loess is Quaternary aeolian loess, which has a highly uniform grain size composition and mainly consists of silt. Comparison of the Ili loess with the loess from the Loess Plateau found that the content of the grain size of 0.05–0.005 mm in the Ili loess is markedly higher than those in the Malan loess from eastern China and the loess from the northern piedmont of the Tianshan Mountains.

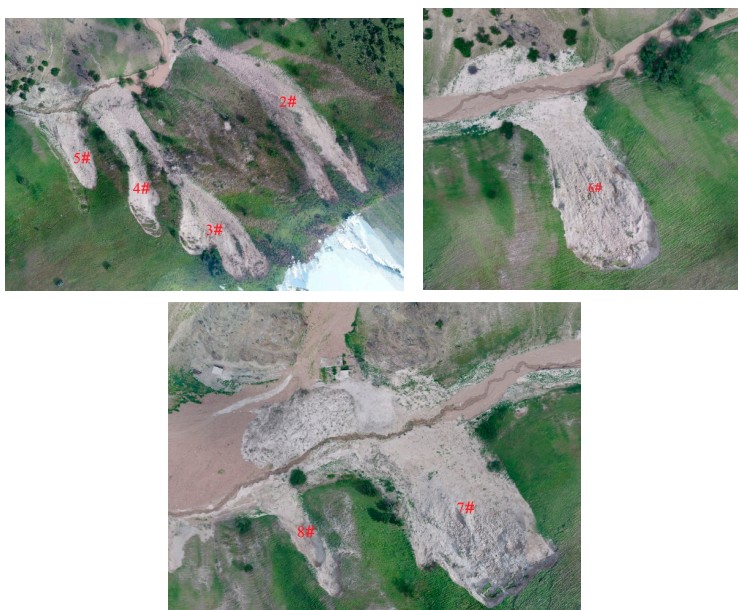

**Figure 7.** The Kezilesai Gully landslide in the Piliqinghe basin.

The landslide that occurred on 24 March 2017 in the Kalayagaqi Village of Kalayagaqi Township was typical of those developed in the Piliqinghe basin. This landslide, referred to as the No. 2 Piliqinghe landslide, blocked the Piliqinghe River, forming a barrier lake (Figure 8). This landslide buried a farmer's fishpond (with an area of 2000 m²), a farmhouse, a small tractor, an excavator, and dozens of fruit trees on the north bank of the Piliqinghe River, with a direct economic loss of more than 200,000 yuan. The Piliqinghe landslide had a clear perimeter and a tongue-shaped pattern in plane, with a gradient of 37° and a main sliding direction of 32° (Figures 9 and 10). The landslide had a width of 96 m, a smallest width of 60 m at the leading edge, a maximum longitudinal length of 175 m, a thickness of 3–6 m, and a total volume of approximately $6.08 \times 10^4$ m³. The landslide was covered with snow approximately 20 cm thick (Figure 11).

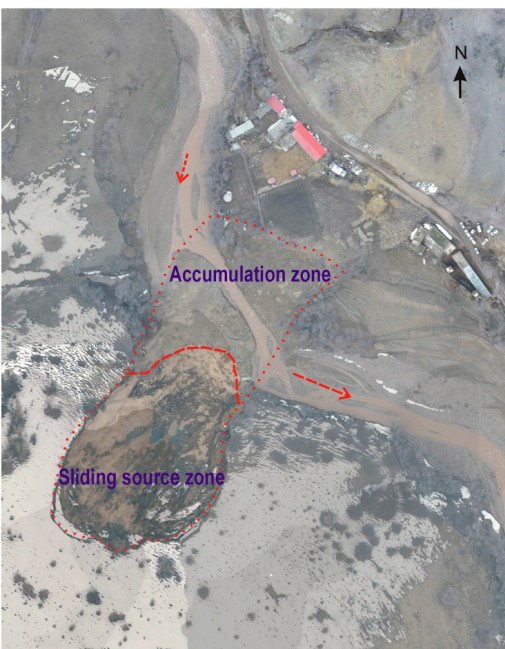

**Figure 8.** Remote sensing image of the No. 2 Piliqinghe landslide.

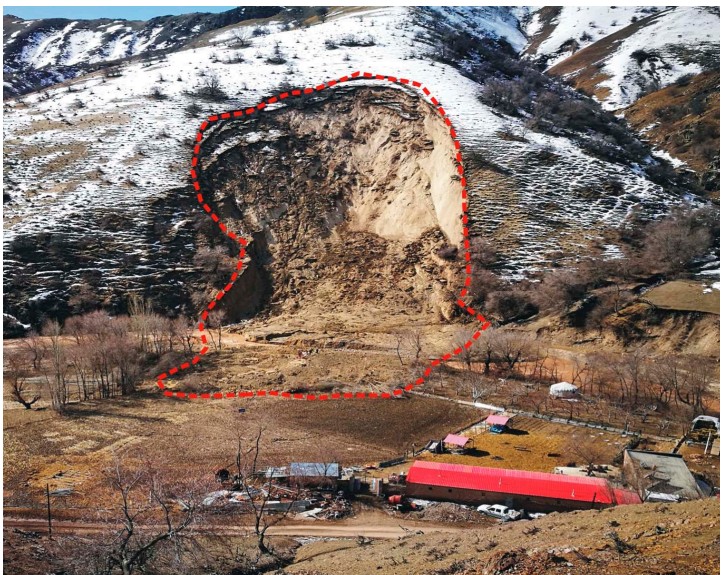

**Figure 9.** Panoramic view of the No. 2 Piliqinghe landslide.

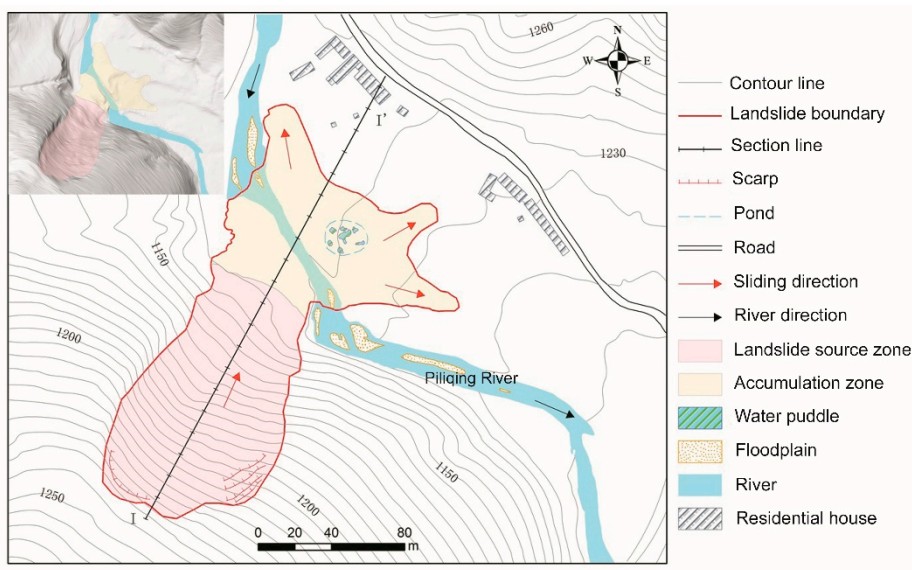

**Figure 10.** Scheme of the engineering geology of the No. 2 Piliqinghe landslide.

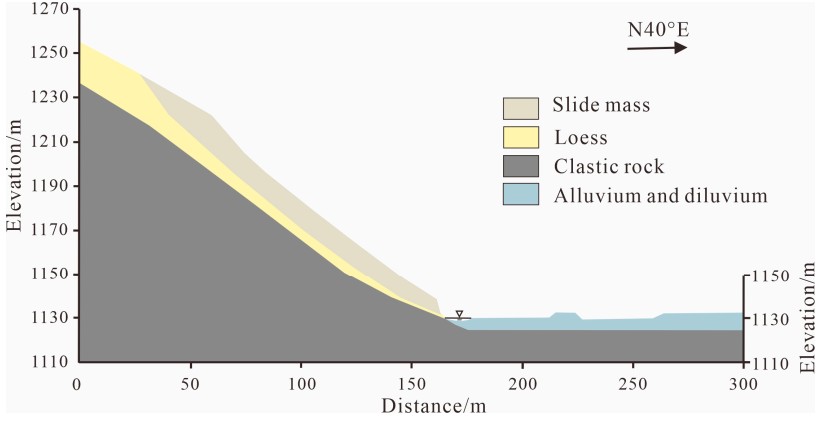

**Figure 11.** Longitudinal profile of the No. 2 Piliqinghe landslide.

As shown in Figure 12, according to the data of the Yining County Meteorological Station, from 1 March to 22 March 2017, the daily minimum and maximum temperatures in the Piliqinghe Basin were below and above 0 °C, respectively, and there was an intense freeze–thaw cycle before the disaster. The rainfall data showed that the total rainfall in the region before the occurrence of the No. 2 Piliqinghe landslide from 10 to 24 March was approximately 20 mm, and there was no rainfall from 19 to 24 March; hence, it was unlikely that rainfall induced the landslide, which occurred on 24 March 2017.

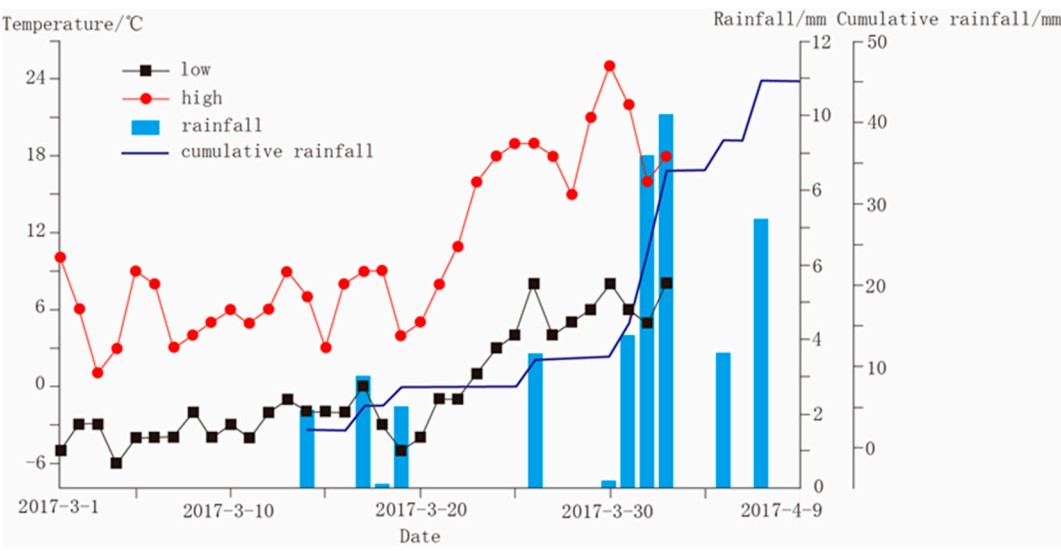

**Figure 12.** Daily minimum and maximum temperatures and rainfall before and after the occurrence of the No. 2 Piliqinghe landslide on 24 March 2017.

Combined with the field investigation results and Zhu Sainan's investigation in this area [32], the results showed that since 1 March 2017, the temperature in this region gradually increased, leading to an increase in snowmelt. As a result, the volume of the Piliqinghe River increased dramatically, the water level rose, and the flow velocity increased. Consequently, the lateral erosion of the leading edge of the landslide on the concave bank was intensified, the strength of the soil at the leading edge decreased, and collapsed occurred, and there was serious scouring of the rock mass at the slope toe, which was the main factor causing the landslide. In addition, as the weather warmed, the snowmelt infiltrated along the cracks in the soil, increasing the sliding force and deadweight of the sliding mass, which was another important factor in the landslide. Therefore, the mechanism of the No. 2 Piliqinghe landslide can be summarized by the following three stages: local collapse at the leading edge → creep sliding and tension cracking of the surface → overall sliding (Figure 13).

(1) Erosion and collapse of the leading edge. In the spring, the snowmelt increased the flow rate of the Piliqinghe River. As a result, the lateral erosion was intensified, and the bank slope was undercut, forming a free face. Therefore, the leading edge of the landslide was partially unbalanced, and multiple sliding collapses occurred, as can be observed in the multi-period remote sensing images.

(2) Creep sliding and tension cracking of the surface. As the snowmelt infiltrated into the slope, the variation in the diurnal temperature around 0 °C led to intense freeze–thaw cycles, which reduced the mechanical strength of the soil, resulting in the formation of cracks on the landslide surface. Meanwhile, as the water content of the soil increased, shear creep occurred on the slope surface, causing creep tension cracking under gravity. Hence, the cracks in the slope were widened, and continuous tensile cracks occurred at the trailing edge.

(3)     Overall deformation and sliding. As the freeze–thaw cycles continued, the frost heave effect further increased the cracks at the trailing edge of the landslide. With the infiltration of a large amount of snowmelt along the cracks, the water content rose sharply, and the soil became nearly saturated. As a result, the friction decreased dramatically, disrupting the original force balance of the loess structure. Consequently, the landslide slid down at high speed, rushing into and burying the fishpond and blocking the Piliqinghe River to form a barrier lake.

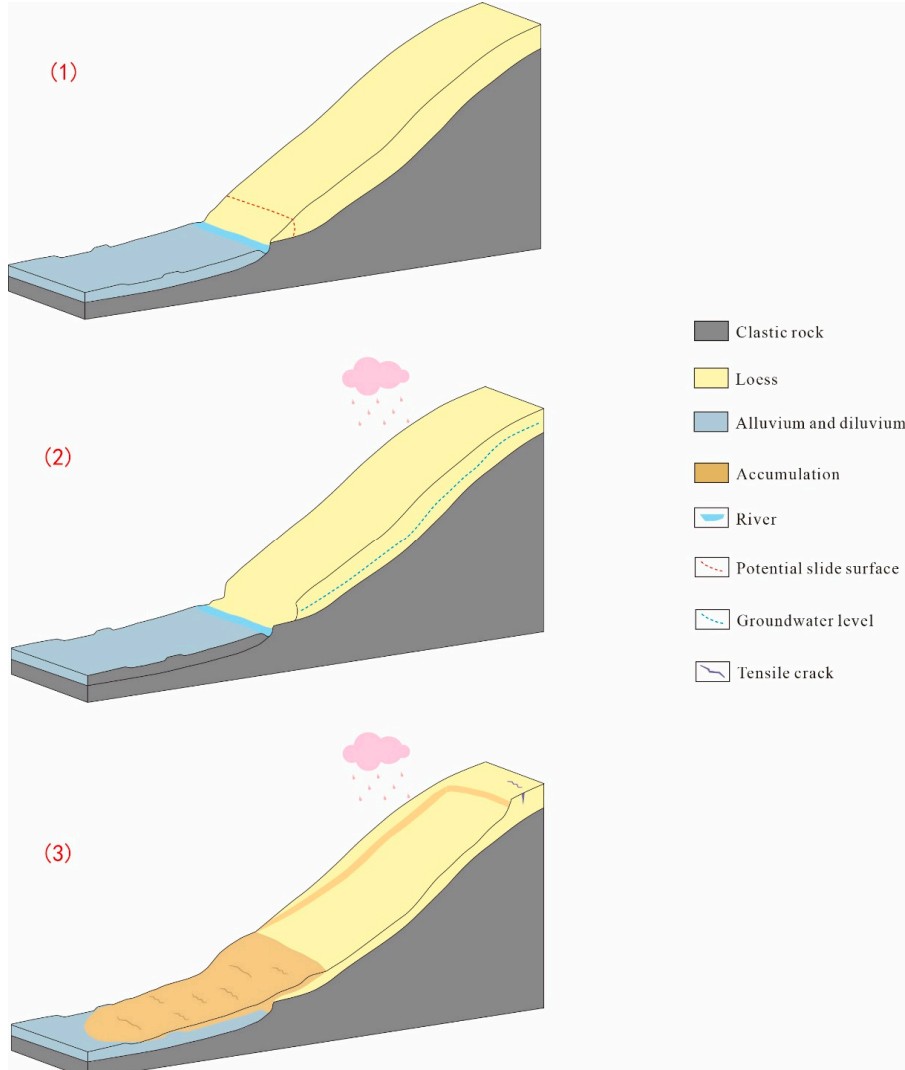

**Figure 13.** Sketch map showing the gestation model of the No. 2 Piliqinghe landslide.

### 3.2. Numerical Simulation of the Landslide

Landslide deformation and failure is a typical large deformation problem, for which Fast Lagrangian Analysis of Continua in 3D (FLAC $^{3D}$) software has been widely used as a numerical simulation tool. This software is based on the finite difference method with a fast Lagrangian analysis algorithm for a continuum and uses an explicit time approximation to obtain the full time-step solution of all equations of motion, thereby allowing tracking of the progressive failure of the slope. For this reason, FLAC $^{3D}$ 5.00was used in the present study to simulate the No. 2 Piliqinghe landslide.

During the computing procedure, the stability factor of slope was computed by the shear strength reduction method. This method has been widely used in slope stability

analysis. Using a series of trial factors of safety $F_s$, the soil strength parameters $c$ and $\varphi$ are reduced as follows:

$$c' = c / F_s \tag{1}$$

$$\varphi' = \arctan\left(\frac{tan\varphi}{F_s}\right). \tag{2}$$

During the numerical computing process, $c'$ and $\varphi'$ are used as the input parameters. The trial factor of safety $F_s$ incrementally increases until the convergent criterion is not satisfied. In this case, the slope is in the limit equilibrium state, and the corresponding $F_s$ is considered as the real factor of safety $F$.

A numerical model (Figure 14) was established based on the engineering geological model (Figure 10). The whole model was meshed using 5078 quadrilateral elements and 68,448 nodes. The default static mode of FLAC was adopted in the simulation. In the model, the displacements in the x-direction on the two sides, all displacements in the y-direction, and the displacement in the z-direction at the bottom were constrained. The surfaces of the landslide model were set as free boundaries. The ideal elastic–plastic constitutive model and the Mohr–Coulomb strength criterion were adopted, and the slope stability factor was calculated using the strength reduction method.

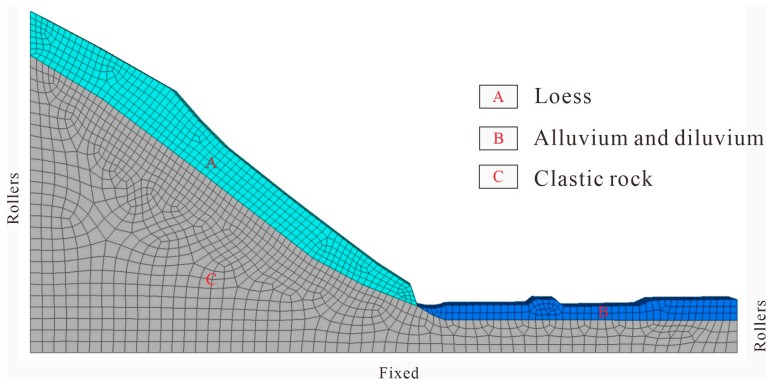

**Figure 14.** A numerical simulation model of the landslide profile.

According to the field investigation data and the model generalization, three different types of rock–soil material were considered in the calculation, namely, loess, alluvium and diluvium, and bedrock. The physico-mechanical parameters of these types of rock–soil and water table level refer to the results of Zhu et al. [32], with the specific parameters listed in Table 2.

**Table 2.** Physical and mechanical parameters of the materials for FLAC $^{3D}$.

| Material Type | Name of Material | Density $\gamma/g*m^{-3}$ | Elastic Modulus E/GPa | Poisson's Ratio v | Tensile Strength t/KPa | Cohesion c/KPa | Internal Friction Angle $\varphi/°$ |
|---|---|---|---|---|---|---|---|
| A | Loess | 1670 (1930) | 0.043 (0.021) | 0.30 (0.32) | 15.72 (13.44) | 17.78 (14.32) | 23.30 (20.08) |
| B | Alluvium & diluvium | 2100 | 0.042 | 0.32 | 13.00 | 13.38 | 21.47 |
| C | Clastic rock | 2542 | 30 | 0.28 | 113.65 | 120 | 45 |

Although the dynamic processes of the slope toe erosion and rainfall infiltration were not simulated in the numerical analysis, the effect of snowmelt on the softening of the slope toe and the influence of rainfall on the soil properties were considered. To this end, the groundwater level was determined according to the results of Zhu et al. [32]; then, the saturated parameters were used for the rock–soil of the slope below the groundwater level in the rainy season, and the saturated parameters were used for all the soil below the river level.

The numerical simulation results for dry and wet season are shown in Figure 15. In the dry season, the plastic shear zone of the slope was distributed at the front of the slope (Figure 15a), and the overall slope stability factor was calculated to be 1.243, indicating that the slope was stable. The shear strain was large at the front of the slope, with a maximum value of $9.61 \times 10^{-4}$, which could be a potential failure surface of the slope toe. Figure 15c shows the contours of the displacement in the landslide, with a maximum value of 0.544 m in the dry season.

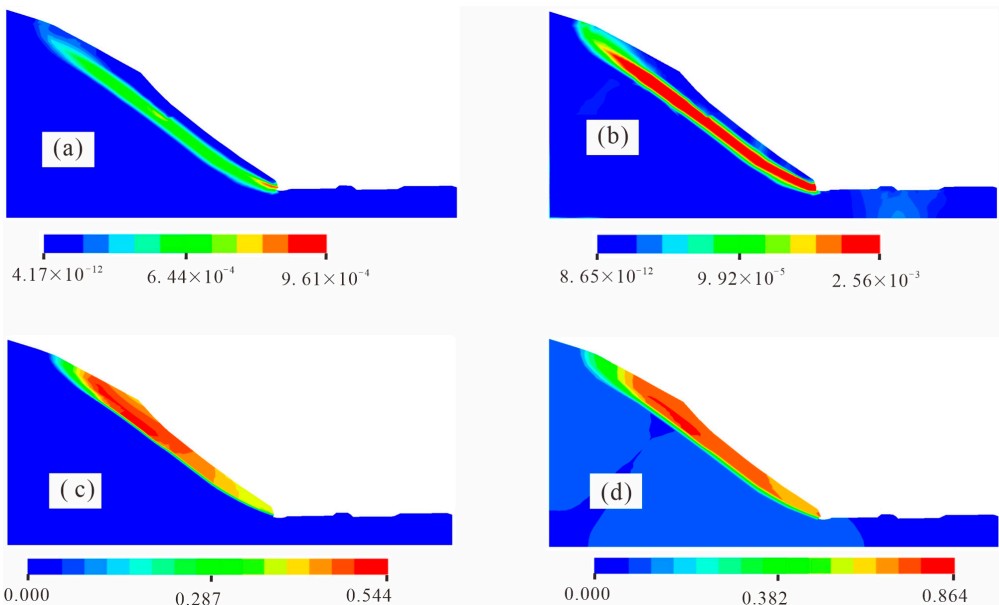

**Figure 15.** Modeling results for dry and wet season: (**a**) contours of the shear strain for the dry season; (**b**) contours of the shear strain for wet season; (**c**) contours of the displacement for the dry season; (**d**) contours of the displacement for the wet season.

In the rainy season, the slope was in the limit equilibrium state, with a calculated stability factor of 0.981. The plastic shear strain, with a maximum value of $2.56 \times 10^{-3}$, was distributed along the sliding zone (Figure 15b), and the trailing edge of the landslide was located at the crack position in the upper part of the slope. In addition, the displacement contours indicate that the maximum displacement in the slope was 0.864 m (Figure 15d). In fact, the cracks at the trailing edge of the Piliqinghe landslide formed a channel for the infiltration of snowmelt and rainfall, resulting in the softening of the soil in the trailing edge of the slope and thus extending the sliding from the back to the middle of the slope. The fluvial erosion and softening of the slope toe also caused the soil of the slope toe to slide. The two factors had an important influence on the progressive failure of the Piliqinghe landslide.

At present, the numerical simulation software cannot simulate the influence of freeze–thaw cycle on landslide stability. How to realize the simulation of the freeze–thaw process can be studied in the future. In this paper, through the adjustment of geotechnical parameters, the softening of slope toe caused by snowmelt infiltration and the influence of rainfall on soil properties are considered, which is a good method for the simulation study of loess landslide under the influence of snowmelt infiltration. For the loess landslide, water is the most important factor. Therefore, under the influence of snowmelt infiltration, the occurrence process of landslide disaster is very complex.

## 4. Conclusions

Based on geological investigation and research on the landslides in the slope region of the Ili Basin in Xinjiang, the present study examined the mode of geological disasters in the region and carried out the engineering geological zoning of the No. 2 Piliqinghe landslide. In addition, a geological structural model of the landslide was built, and its deformation and failure process was simulated using the numerical software FLAC $^{3D}$. The following conclusions were drawn:

(1) The disaster process of the No. 2 Piliqinghe landslide was as follows: fluvial scouring of the slope toe leading to the local sliding of the leading edge → snowmelt infiltration resulting in the creep sliding and tension cracking of the slope surface → overall sliding of the slope.

(2) The infiltration of snowmelt due to temperature increase was the direct cause of landslide formation. The infiltrated snowmelt was transformed into groundwater, which saturated the loess and increased the pore water pressure, thereby decreasing the slope stability factor and leading to an overall sliding under gravity.

(3) The stability and the underlying deformation and failure mechanism of the No. 2 Piliqinghe landslide were investigated using the finite difference strength reduction method. The results showed the following. In the dry season, the slope had a stability factor of 1.254, indicating that it was stable. The shear strain at the front of the slope was large, with a maximum value of $3.59 \times 10^{-4}$, and the maximum displacement in the slope was 0.304 m. In the rainy season, the slope was in the limit equilibrium state, and the stability factor was calculated to be 0.985. The plastic shear strain, which had a maximum value of $9.29 \times 10^{-4}$, was distributed along the sliding zone, and the trailing edge of the landslide was consistent with the crack position in the upper part of the slope. The maximum displacement in the slope was 0.636 m. Therefore, regarding the prevention of loess landslide disasters in the Ili region, the results suggest that cracks in the trailing edge of the slope have an extremely important influence on the landslide and that additionally, instability in the middle and back parts of the landslide has a strong impact on the stability of the whole slope and should be avoided. These findings can be used as a reference for the analysis of the disaster mechanism and movement characteristics of similar loess landslides and provide further theoretical support for disaster prevention and mitigation.

**Author Contributions:** Methodology, M.Z. and W.G.; Field Investigation, M.Z., H.S., S.Z. and Y.W.; Software, M.Z., T.Z. and W.G.; Data curation, M.Z.; Writing—Original Draft Preparation, M.Z. and W.G.; Writing—Review and Editing, M.Z., R.H. and T.Z. All authors have read and agreed to the published version of the manuscript.

**Funding:** We appreciate financial support by the National Natural Science Foundation of China (NSFC) (No. 41671355); Geological Survey Project (No. DD20179609); The Chinese Academy of Sciences (CAS) "Light of West China" Program; Yan'an University Project (YDBK2019-37). Grateful appreciation is expressed for the support.

**Data Availability Statement:** Data sharing not applicable.

**Acknowledgments:** This authors are grateful to editors and reviewers for their enthusiastic help and valuable comments.

**Conflicts of Interest:** The authors declare no conflict of interest.

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
