# Peer review of "Mechanistic Investigation of Typical Loess Landslide Disasters in Ili Basin, Xinjiang, China"

_sustainability, doi:10.3390/su13020635_

Round 1
Reviewer 1 Report
This is a nice research outcome and timely paper within the scope of the journal. Also, the work has lots of merits for future scholars. However, I have few comments:
- the introduction started immediately with the Chinese experience. I would recommend that the authors bring few examples from other countries where landslides are always happening. This will improve the understanding of international readers in their own areas.
- Line 67 --- many land-slide triggering factors....you should at least bring few of them. otherwise the sentence is vague.
- You might find few interesting findings from Alaska and Alberta about the similar events and you may want to take a look if they have built similar models.
- Why have you chosen this Ili region in particular? This fact is somewhat not compact in your introductory writing.
- I will suggest that the authors prepare a schematic diagram of the adopted methods in their research after Figure 1 on page 4. This will help readers to know the method in a nut shell.
- Remotely sensed image on Fig 5. If I am not mistaken, you probably did not mention to specification of the image in the earlier section! If this images are freely available, it will encourage more researchers to work with this issue.
- Figure 11 and 12 can be merged together to understand the basic rainfall and temperature data in a same figure.
- Numerical simulation looks nice. However, the discussion should be carried out to demonstrate that in this particular geographical setting, what were the major challenges to be considered before numerical modeling? this part is simply missing. I will suggest, the authors write a section before conclusion about the uniqueness of their work and how the scope may enhance understanding the similar geographic location in the world.
- conclusion looks nice.
Author Response
Dear reviewer
Thank you for reviewing our article our manuscript entitled “Mechanistic investigation of typical loess landslide disasters in Ili Basin, Xinjiang, China”. (ID: sustainability-1060552). Those comments are valuable and very helpful to further improve our manuscript. We have carefully evaluated the critical comments and thoughtful suggestions, responded to these suggestions point by point, and revised the manuscript accordingly. We hope that the modifications would meet with approval. All changes made to the text are in red so that they may be easily identified.
Attached is the our response to the your comments, please see the attachment.
Thank you and best regards.

Reviewer 2 Report
In all this was an interesting paper that was well structured and presented.
There were two areas that I would recommend for improvement:
1: It was unclear whether the outline in lines 237-267 were a summary of the analytical conclusions or whether they were the summary of the fieldwork. If the former perhaps they should explain that and avoid duplication of summary results later on. If they are the result of the fieldwork, what observations were taken in the field that lead to these conclusions?
2. It was frustrating trying to locate the areas mentioned in the text. It would help to set the scene if map 1 had the rivers marked on it or some sample contours, for example. Again in figure 4, I struggled to work out where the map was of, is it possible to mark the location in map 1, perhaps or to give a grid reference for one of the towns mentioned, for example Piliqinghe?
I also have one minor editorial point:
On the whole the language and formatting was excellent but there was one tweak required in Table 1 where the years in the left-hand column were wrapped around more than one line, making it a bit difficult to follow.
Author Response

(The authors gave the same response as above.)

Reviewer 3 Report
The following parts must be modified.
- The watertable is not clearly shown in Figure 15 for both seasons. It is important in the numerical simulations.
- In Figure 15, the LHS boundary for dry season is not large enough which should be extended and simulated again. Therefore, the possible failure mechanism can be obtained properly.
- In p.15, stability factor is mentioned in the content. Do the authors define stability factor? It is not commonly used terminology in slope stability analyses.
- Line 337, finite element strength reduction method? It differs from the previous content.
Author Response
Dear reviewer
Thank you for reviewing our article our manuscript entitled “Mechanistic investigation of typical loess landslide disasters in Ili Basin, Xinjiang, China”. (ID: sustainability-1060552). Those comments are valuable and very helpful to further improve our manuscript. We have carefully evaluated the critical comments and thoughtful suggestions, responded to these suggestions point by point, and revised the manuscript accordingly. We hope that the modifications would meet with approval. All changes made to the text are in red so that they may be easily identified.
Attached is the our response to the your comments, please see the attachment.
Thank you and best regards.
Yours sincerely,
Wen-wei Gao & ~co-authors.
E-mail: gaowwan@163.com
